# Pathophysiology of Immune Checkpoint Inhibitor-Induced Myocarditis

**DOI:** 10.3390/cancers14184494

**Published:** 2022-09-16

**Authors:** Rosa Jiménez-Alejandre, Ignacio Ruiz-Fernández, Pilar Martín

**Affiliations:** 1Centro Nacional de Investigaciones Cardiovasculares (CNIC), 28029 Madrid, Spain; 2CIBER de Enfermedades Cardiovasculares (CIBER-CV), 28029 Madrid, Spain

**Keywords:** myocarditis, ICI-myocarditis, immunotherapy, irAEs, cancer therapy, cardiotoxicity, T cells, anti-PD-1, anti-CTLA-4

## Abstract

**Simple Summary:**

Myocarditis is an infrequent but highly hazardous complication of the cancer therapy of immune checkpoint inhibitors (ICIs). The study of the pathophysiology of this disease is an active field of research and a clearer comprehension of the mechanisms is crucial to provide an accurate diagnosis, appropriate therapy, and to prevent cardiac adverse toxicities occurring during ICI treatment that compromise the continuation of the cancer treatment. This review provides an update of the currently approved ICIs and their relationship with myocarditis induction through boosting the immune system. It also discusses preclinical models of ICI-associated myocarditis and their contribution to the state of the art and presents recent advances in the pathogenesis of the disease.

**Abstract:**

Immune checkpoint inhibitors (ICIs) have recently emerged as strong therapies for a broad spectrum of cancers being the first-line treatment for many of them, even improving the prognosis of malignancies that were considered untreatable. This therapy is based on the administration of monoclonal antibodies targeting inhibitory T-cell receptors, which boost the immune system and prevent immune evasion. However, non-specific T-cell de-repression can result in a wide variety of immune-related adverse events (irAEs), including gastrointestinal, endocrine, and dermatologic, with a smaller proportion of these having the potential for fatal outcomes such as neurotoxicity, pulmonary toxicity, and cardiotoxicity. In recent years, alarm has been raised about cardiotoxicity as it has the highest mortality rate when myocarditis develops. However, due to the difficulty in diagnosing this cardiac condition and the lack of clinical guidelines for the management of cardiovascular disease in patients on therapy with ICIs, early detection of myocarditis has become a challenge in these patients. In this review we outline the mechanisms of tolerance by which this fatal cardiomyopathy may develop in selected cancer patients treated with ICIs, summarize preclinical models of the disease that will allow the development of more accurate strategies for its detection and treatment, and discuss the challenges in the future to decrease the risks of its development with better decision making in susceptible patients.

## 1. Introduction

Immune cell co-inhibitory molecules such as cytotoxic T-lymphocyte-associated antigen 4 (CTLA-4), programmed cell death 1 (PD-1) and its ligand programmed cell death-ligand 1 (PD-L1), are membrane receptors that serve as regulators to prevent uncontrolled T cell responses occurring in an organism [1,2]. Given their mechanism of action, they have been exploited as therapeutic targets for cancer immunotherapy to boost the immune system and overcome immune evasion by cancer cells [3]. Immune checkpoint inhibitors (ICIs) are monoclonal antibodies aimed at functionally targeting these regulatory receptors and thereby enhancing T-cell activation and activity after blockade [2]. These have been highly successful in the clinic and an increasing number of ICIs have been approved for the treatment of a wide variety of tumors in the last decade (Table 1).

However, unspecific unleashing of immune system results in a wide variety of immune-related adverse events (irAEs) [4,5]. Among the myriad of toxicities, myocarditis is an uncommon (0.04–1.14%) but relevant event, due to its high lethality (20–50%) [6,7,8,9,10,11,12]. ICI-related myocarditis has a low incidence, complex diagnosis, and few studied cases, making it difficult to reach a consensus on an efficient way to approach this pathology, especially with a very heterogeneous clinical presentation. On this note, this review aims to provide an overview of the available knowledge about the mechanisms leading to the loss of tolerance during ICI-induced myocarditis, the available preclinical models that recapitulate the pathology for study, and the latest avenues of research working towards a deeper comprehension of this heart condition to develop effective diagnostic and therapeutic tools.

## 2. Immune Tolerance

T cells are a heterogeneous subset of immune cells pivotal for the development, maintenance, and self-control of adaptive immune responses. They are characterized by the expression of a unique T cell receptor (TCR), key for the establishment of a tailored immune response against a wide repertoire of antigens. After generation and maturation in the bone marrow and thymus, naïve CD4^+^ and CD8^+^ T cells leave this organ and are dependent on antigen-presenting cells (APCs) to be primed and become T helper cells (Th) or cytotoxic T lymphocytes (CTLs), respectively. To achieve this, T cells respond to a two-signal activation model: 1) induction of the CD3 signaling pathway upon TCR binding an antigen displayed on the major histocompatibility complex (MHC), and 2) co-stimulation through CD28 binding to CD80/CD86 molecules [13]. This model enables only T cells provided with both signals to become activated, proliferate, and elicit their function.

One of the major evolutionary challenges of the immune system is to exert efficient immune responses against given pathogenic agents that jeopardize homeostasis while also being constantly exposed to self and non-harmful antigens without eliciting a detrimental response against them. Optimization of the immune system to protect self-tissues has led to the development of immune tolerance mechanisms.

Central tolerance is the process originating in the thymus, where apoptosis is induced in T cell clones carrying a receptor capable of recognizing self-antigens with high affinity. In this process, termed negative selection, immature thymocytes are exposed to a wide range of circulating and tissue-specific antigens (peptides) by APCs and medullary thymic epithelial cells (mTECs), respectively [14]. Negative selection drastically reduces the potential number of autoreactive T cells released into the circulation, however, 24–40% of them manage to escape selection due to the incomplete representation of the self-peptidome [15], hence peripheral tolerance mechanisms have emerged to prevent activation of these self-reactive T cell clones.

The first and most structural mechanism of peripheral tolerance is the restriction of T cell access to the circulation through blood vessels and secondary lymphoid organs, sequestering most priming antigens in immune privileged sites where access is only allowed in inflammatory settings and T cell activation is relegated to the encounter with APCs [1,16].

Another mechanism of peripheral tolerance is the functional inactivation of T cells occurring in the absence of costimulatory signals on the surface of tissue target cells and immature APCs. Thus, when autoreactive T cells recognize a peptide displayed on any of these, an unresponsive behavior called anergy is induced due to a lack of the second signal. The tolerogenic TCR stimulation leads to activation of the transcription factor NFAT1 which in turn promotes the expression of proteins inducing this anergic state [17]. This anergic state is maintained even upon successive antigen presentation when both signals are present, but it can be reversed in the long term [18].

Autoreactive T cells can also be controlled by the T regulatory (Treg) cell population. This subset is generated from thymocytes with high affinity for self-peptides where Foxp3 transcription factor expression is induced (natural or thymus-derived Treg cells; tTreg) or in the periphery after TCR stimulation with a specific combination of cytokines including transforming growth factor (TGF)-β and interleukin (IL)-2 (adaptive, induced or peripherally derived Tregs; pTreg) [19,20,21]. Further, anergic T cells are able to upregulate Foxp3 and differentiate into this subset of cells [22]. The Treg population helps to prevent or suppress the activation of self-reactive T cells via the production of anti-inflammatory cytokines, competition for stimulatory factors such as IL-2 and the expression of coinhibitory receptors in their membrane [14,21]. The latter are also referred to as immune checkpoints and they are immunomodulatory membrane receptors that interfere with the priming or activity of T cells by directly competing or impeding the binding of T cells to APCs or their target cells, respectively. These are upregulated as a result of T cell activation and act as a homeostatic mechanism to control the immune responses and ensure a termination, while also preventing autoimmunity [2]. Immune checkpoint molecules have gained attention for their use in cancer therapies as their blockade prevents cancer immune evasion (Figure 1).

CTLA-4 or CD152 is an Immune Checkpoint molecule that belongs to the CD28 Ig superfamily and presents a similar structure to this molecule. CTLA-4 can capture CD80 and CD86 (also named B7.1 and B7.2) on APCs with higher affinity than performed by CD28, blocking the costimulatory signaling cascade and activation of T cells [23]. CTLA-4 molecule is found expressed on the membrane of Treg cells, as one of their mechanisms of tolerance induction [24]. It is also upregulated upon activation on other CD4+ T cell subsets to restrain exacerbated activation and proliferation by hijacking costimulatory molecules on APCs [25] while also signaling through its association with the phosphatases SHP-2 and PP2A, inhibiting signaling cascades downstream of TCR and CD28 [26,27].

Other important immunomodulatory molecules are PD-1 (CD279) and its ligands PD-L1 (CD274) and PD-L2 (CD273). PD-1 is another member of the CD28 Ig superfamily and is expressed by T cells during activation, B cells, natural killer, and myeloid cells. PD- L1 can be found on APCs, T and B cells, the thymic cortex and a number of non-haematopoietic lineages including vascular and cardiac endothelial cells; and PD-L2 is expressed by dendritic cells, macrophages, some B cells, mast cells, Th2 cells, and the thymic medulla. Additionally, PD-1 and its ligands are expressed by a number of cancer cell lineages as a mechanism to promote tumoral immune evasion [28]. The recognition of its ligand prompts apoptosis in the PD-1-bearing cells by recruiting SHP-2 phosphatases, providing a mechanism to end immune responses by unwanted targets or regulatory cell types [29].

Likewise, other molecules present similar mechanisms of inhibition in the immune synapse. The lymphocyte activation gene-3 (LAG-3) (CD223) is another immune checkpoint molecule that functions as a negative regulator of the immune response upon binding to MHC class II molecules. This interaction limits T cell activation, proliferation, and cytokine secretion. It appears expressed on activated T cells, some natural killer cells, B cells and plasmacytoid dendritic cells [30]. T cell immunoglobulin and mucin domain-containing protein 3 (TIM-3) is expressed on T cells, B cells, NK cells, DCs, monocytes and macrophages, and induces inhibitory and apoptotic pathways when it interacts with one of its ligands: galectin-9, carcinoembryonic antigen cell adhesion molecule 1 (Ceacam-1), high-mobility group protein B1 (HMGB1) and phosphatidyl serine (PtdSer) [31]. CD69 has been for years envisioned just as an early activation marker but in the last decade its important role as co-inhibitor receptor modulating the Treg/Th17 balance has emerged [32,33]. T cell immunoglobulin and immunoreceptor tyrosine-based inhibitory motif [ITIM] domain (TIGIT) and V-domain Ig suppressor of T cell activation (VISTA) are also involved in blocking the activation and differentiation of T cells and the production of cytokines [31].

## 3. Immune Checkpoint Inhibitors in Cancer

Cancer presents a global concern and several approaches have been designed to target the different presentations of this pathology. Blockade of immune checkpoint molecules has been a major breakthrough since it has increased the survival of many cancer patients by unprecedented numbers in recent decades. One of the hallmarks that cells acquire to develop malignant behavior is the evasion of immune mechanisms established to avoid tumor growth, including the upregulation of these molecules to prevent destruction by tumor-specific responses [3].The growing use of ICIs in the clinics to boost the immune system have demonstrated a great efficacy against tumors, but it has also presented immunotherapy as a “double-edged sword”, in which immune overactivation unleashes autoimmune responses. These irAEs comprise a myriad of toxicities that can occur against any tissue. The reported incidence of irAEs varies depending on the targeted molecule, but they are experienced in about 60% to 90% of patients treated with ICIs [34] and the most commonly affected tissues are the skin, gastrointestinal track, and liver. Toxicities emerge between 2 and 16 weeks after treatment initiation on average [4,34]. Although most symptoms are mild, irAEs have an overall associated mortality rate of 0.6% and the risk increases to 1.23% for combination therapy [35]. Most irAEs-associated deaths are induced by colitis in patients treated with anti-CTLA-4, while pneumonitis, hepatitis, and neurotoxicity are the main cause of death in patients in whom the PD-1/PD-L1 axis is inhibited [12]. The differential effect when targeting different pathways may be explained due to the nature of the immune checkpoint molecules themselves, their specific role in the pathophysiology of the disease and the kinetics of the immune response—whereas CTLA-4 blockade acts unleashing of T cells, the anti-PD-1/PD-L1 axis acts at a later point preventing suppression of exhausted or autoreactive cells. Additionally, tissue-related factor may also contribute to differential toxicities upon ICI treatment. For example, hypophysitis is more common with anti-CTLA-4 and this molecule is found expressed in the pituitary [36] while PD-1 ligand has been found expressed in endothelial cardiac cells [37], which could contribute to heart-related diseases. Nevertheless, further research is needed to elucidate the mechanisms and understand the cause of distinct therapy adverse effects. In the case of combination therapy, colitis and myocarditis are responsible for most deaths. Among all adverse events caused by ICIs, myocarditis (ICI-myocarditis) is the one having the highest mortality [12].

## 4. Myocarditis Induced by Immune Checkpoint Blockade

Myocarditis is one of the adverse effects observed upon ICI treatment. This disease is characterized by myocardial inflammation and the infiltration of immune cells into the cardiac tissue. Although it is not one of the most prevalent adverse effects, with an estimated incidence of 0.04–1.14% [6,9], it has the highest mortality rates among all irAEs (20–50%) [7,10,11,12]. Our current knowledge of the epidemiology and clinical presentation of ICI-associated myocarditis is mostly derived from retrospective studies or case reports. However, prospective studies have recently being designed to better understand the relationships between ICI therapy and cardiac adverse events (Table 2).

Clinical presentation of ICI-derived myocarditis is very heterogeneous, ranging from asymptomatic to life-threatening. Most severe cases develop during the first 6 weeks after initiation of therapy [9] and may present with chest pain, dyspnea, elevated troponin and creatine kinase levels, ST-segment elevation, cardiac arrhythmia, and other electrocardiographic abnormalities [38]. These features can be easily detected, but none are specific to myocarditis and overlap with other cardiac pathologies. The gold standard technique for the differential diagnosis of myocarditis is endomyocardial biopsy, which allows direct detection of inflammatory infiltrates in the tissue [39]. The disadvantage of this technique is its invasiveness and potential complications, as well as its low sensitivity due to the patchy pattern of myocarditis, which makes it difficult to collect the ideal sample from the affected tissue. Cardiovascular magnetic resonance (CMR) has arisen as a promising non-invasive alternative to detect myocardial edema and non-ischemic myocardial injury secondary to inflammation following the updated Lake Louis criteria. However, this technique is not always available in all healthcare facilities [10,40].

Combining cases misdiagnosed as other pathologies with asymptomatic patients (a common condition in myocarditis), ICI-myocarditis could be underdiagnosed. Recently, several guidelines from different organizations including the National Comprehensive Cancer Network (NCCC), the American Society for Clinical Oncology (ASCO), and the European Society for Medical Oncology (ESMO) and the Society for Immunotherapy of Cancer (SITC) have been released in an effort to reach a consensus in the diagnosis and management of ICI-related toxicities [41,42,43,44]. Several factors have been linked with increased risk of severe cases and cardiac events during ICI therapy in contrast to asymptomatic or mild cases [7,44]. Among these, we can find popular cardiovascular risk factors such as age, smoking, pre-existing diabetes mellitus or hypertension. Troponin levels are worth discussing since the appearance of major adverse cardiac events is significantly related to admission, peak, and final troponin T levels [7]. This factor is widely used as a cardiac damage biomarker and included in the current guidelines to stratify the different cases regarding its severity [41]. Elevated troponins have been reported in cancer patients related to the worsening of the oncological disease and in other conditions commonly suffered by these patients, such as anemia or sepsis. Additionally, other irAEs can also lead to a troponin rise [45]. Still, although its predictive use can be beneficial in combination with other readouts, it provides no sensitivity nor specificity and caution is required especially regarding false positive cases [46]. Waliany et al. acknowledge this matter in a study where they proposed a high-sensitivity troponin I (HsTnI) threshold of 55 ng/L, corresponding to the 99th percentile for the general population [47]. However, they also argued that a higher threshold could achieve better positive predictive values and further studies with bigger cohorts would be needed to establish this value.

Recently, a new circulating microRNA was described as a specific biomarker for the detection of acute myocarditis [48], but it is unknown whether it could also be a biomarker for the detection of ICI-myocarditis. In current clinical practice, if there is no clinical evidence of myocarditis, none of the above tests are performed, so cases of subclinical myocarditis may be underestimated [7].

Meta-analysis of the reported cases during the last few years revealed an increased myocarditis occurrence over time [49], evidencing growing awareness of this pathology in the clinics. Nevertheless, there remains a clear need for more specific, accessible, and sensitive diagnostic tools to address all presentations of this pathology.

Data associating the different types of ICIs with myocarditis are very scarce. Anti-PD-1 antibody therapy leads to more cases of cardiac adverse effects than anti-CTLA-4 therapy [49,50] and several reports and meta-analyses studies have reported increased cases and worse prognosis of myocarditis patients treated with ICIs in combination compared to monotherapies [7,11,49,50]. However, a meta-analysis integrating data from randomized clinical trials revealed no significant differences between the incidence from both groups [51], so further research is required to establish the synergistic effects of ICIs.

Regarding treatment, current guidelines recommend discontinuation of immunotherapy in severe cases and administration of intravenous methylprednisolone or prednisone [41,42,43,44]. Additionally, other immunosuppressive treatments such as anti-thymocyte globulin (ATG), infliximab, immunoglobulin (IVIG), abatacept or mycophenolate are suggested for steroid-resistant cases. Although mitigation of toxicities is effective, there is a lack of targeted therapeutic approaches that do not worsen the cancer status and may allow ICI resumption. A better understanding of the pathophysiology of ICI-myocarditis could reveal new therapeutic targets to combat it more specifically and on considering the possibility of not having to discontinue anti-tumor treatment.

**Table 2 cancers-14-04494-t002:** Studies reporting cases of ICI-associated myocarditis.

Type of Study	ICI Treatment Used	ICI-Associated Myocarditis Reported Cases	Total Number of ICI-Treated Patients Studied	Reference
Retrospective (VigiBase database)	Nivolumab (anti-PD-1) Pembrolizumab (anti-PD-1) Ipilimumab (anti-CTLA-4) Atezolizumab (anti-PD-L1) Durvalumab (anti-PD-L1) Avelumab (anti-PD-L1) Combination anti-PD-1/PD-L1 + anti-CTLA-4	101	101	[11]
Retrospective (VigiBase database)	Nivolumab (anti-PD-1) Pembrolizumab (anti-PD-1) Ipilimumab (anti-CTLA-4) Atezolizumab (anti-PD-L1) Durvalumab (anti-PD-L1) Avelumab (anti-PD-L1) Combination anti-PD-1 (Nivolumab or Pembrolizumab) + anti-CTLA-4	122	31,321	[49]
Both retrospective and prospective (8 center American registry)	Nivolumab (anti-PD-1) Pembrolizumab (anti-PD-1) Ipilimumab (anti-CTLA-4) Tremelimumab (anti-CTLA4) Atezolizumab (anti-PD-L1) Durvalumab (anti-PD-L1) Avelumab (anti-PD-L1) Combination anti-PD-1 (Nivolumab or Pembrolizumab) + anti-CTLA-4 (Ipilimumab) Combination anti-PDL1PD-L1 (avelumab or durvalumab) + anti-CTLA-4 Tremelimumab (anti-CTLA-4)	35	964	[7]
Prospective (19 center international registry)	Nivolumab (anti-PD-1) Pembrolizumab (anti-PD-1) Ipilimumab (anti-CTLA-4) Tremelimumab (anti-CTLA4) Atezolizumab (anti-PD-L1) Durvalumab (anti-PD-L1) Avelumab (anti-PD-L1) Combination anti-PD-1 (Nivolumab or Pembrolizumab) + anti-CTLA-4 (Ipilimumab) Combination anti-PDL1PD-L1 (avelumab or durvalumab) + anti-CTLA-4 Tremelimumab (anti-CTLA-4)	113	3637	[52]
Prospective (23 center international registry)	Nivolumab (anti-PD-1) Pembrolizumab (anti-PD-1) Ipilimumab (anti-CTLA-4) Tremelimumab (anti-CTLA-4) Atezolizumab (anti-PD-L1) Avelumab (anti-PD-L1) Combination anti-PD-1/PD-L1 + anti-CTLA-4	103	103	[10]
Retrospective (Massachusetts General Hospital database)	Nivolumab (anti-PD-1) Pembrolizumab (anti-PD-1) Combination Ipilimumab (anti-CTLA-4) + Nivolumab (anti-PD-1)	10	10	[53]
Prospective (Danish Registry)	Nivolumab (anti-PD-1) Pembrolizumab (anti-PD-1) Ipilimumab (anti-CTLA-4)	11	1103	[54]
Retrospective (RPCCC medical records)	Nivolumab (anti-PD-1) Pembrolizumab (anti-PD-1) Atezolizumab (anti-PD-L1)	23	23	[55]
Retrospective (IBM MarketScan research databases)	Nivolumab (anti-PD-1) Pembrolizumab (anti-PD-1) Ipilimumab (anti-CTLA-4) Atezolizumab (anti-PD-L1) Avelumab (anti-PD-L1) Durvalumab (anti-PD-L1) Combination anti-PD-1 + anti-CTLA-4	6	12,187	[56]
Retrospective (University of Tsukuba Hospital records)	Nivolumab (anti-PD-1) Pembrolizumab (anti-PD-1) Durvalumab (anti-PD-L1)	4	625	[57]
Clinical trial	Combination nivolumab (anti-PD-1) + relatlimab (anti-LAG-3)	6	355	[58]

## 5. Preclinical Models of ICI-Myocarditis

The use of preclinical models has not only provided the main source of knowledge about the anti-tumor activity of ICIs, but also provides insight into the cells and molecular players responsible for the associated adverse toxicities, such as myocarditis. These models are based on the genetic deletion of the immune checkpoint molecules or their blockade by monoclonal antibodies for a given period of time.

### 5.1. Genetic Deletion of Immune Checkpoint Molecules Causes Myocarditis in Preclinical Models

#### 5.1.1. Pdcd1 Knockout Mice

It seems clear that genetic susceptibility is a risk factor for the development of ICI-derived myocarditis during cancer treatment as substantial individual variation has been reported and some individuals seem to have a predisposition to autoimmunity. However, the variations in the genome that predispose to the disease have not been fully elucidated. Similarly, PD-1 deficient mice (*Pdcd1*^−/−^) develop autoimmune diseases affecting different tissues depending on the genetic background. On the one hand, C57BL/6 and non-obese diabetic (NOD) PD-1 knockout (KO) mice have no cardiovascular disease but suffer nephritis and arthritis, and Type I diabetes, respectively [59,60]. However, BALB/c PD-1 KO mice, but not PD-1 RAG2 double-KO, present dilated cardiomyopathy and die prematurely due to congestive heart failure. Unlike ICI-derived myocarditis in cancer patients, immune infiltration was not observed in PD-1 KO mice hearts, but rather circulating autoantibodies against cardiac troponin are the cause of the phenotype observed. This can be reproduced in BALB/c wild type (WT) mice by injection of a monoclonal antibody against cardiac troponin [61,62]. Murphy Roths large (MRL) PD-1-deficient mice also present circulating autoantibodies, but against cardiac myosin. As a result, they suffer fulminant myocarditis with massive cardiac infiltration of CD4^+^ and CD8^+^ T cells and myeloid cells. In these mice, infiltrating T cells are only found activated in the heart but not in lymphoid organs, suggesting that ICI-derived myocarditis is mediated by an antigen-specific autoimmune response [63].

Specific contribution of CD8^+^ and CD4^+^ T cells in ICI-myocarditis have also been studied in *Pdcd1*^−/−^ mice. Tarrio et al. observed in a model of experimental cytotoxic CD8^+^ T lymphocyte-mediated myocarditis that transferred PD-1-deficient CD8^+^ T cells were capable of inducing stronger responses against the heart than WT CD8^+^ T cells. Mice injected with PD-1 KO-derived CD8^+^ T cells showed more immune cells in heart-draining lymph nodes, mostly CD11c^+^ cells and CD8^+^ Interferon-γ^+^ (IFNγ) cells. Hearts also presented more leukocyte infiltration and enhanced recruitment of neutrophils and macrophages compared to WT CD8^+^-transferred mice. The mechanisms by which PD-1 deficiency enhanced myocardial inflammation in this model were explained by authors as an increased proliferative capacity and cytotoxicity in CD8^+^ T cells lacking PD-1, with higher secretion of IFNγ and granzyme B and less IL-10. In the same article, Tarrio et al. induced experimental autoimmune myocarditis (EAM) in BALB/c WT, PD-1 and PD-L1 KO mice. EAM is a CD4^+^ T cell-dependent myocarditis model consisting of the injection of an immunogenic fragment of the α-myosin heavy chain peptide (MyHCα 614–629) emulsified with complete Freund adjuvant. They found an increased susceptibility to EAM in PD-1 KO mice with increased expression of IL-17A, IFN-γ, and the transcription factor RORγt in the heart of PD-1-deficient mice and more cardiac infiltration of T cells (mainly CD4^+^), macrophages, and high number of neutrophils [64].

#### 5.1.2. Pdcd1 Ligand 1 Knockout Mice

As mentioned above, PD-L1 expression is not limited to immune cells, but it is also overexpressed in self tissues and several tumor cell types, inducing immune suppression. In the context of ICI-myocarditis, the induction of PD-L1 expression on cardiac endothelial cells under inflammatory conditions is especially important and it has been observed in mice and humans [64,65].

Spontaneous myocarditis co-occurring with pneumonitis has been reported in mouse models of PD-L1 knockout in MRL and MRL^−*Faslpr*^ background [66]. These mice die at an early age due to congestive heart failure and have enlarged hearts with mononuclear infiltrates consisting of macrophages, CD8^+^ and CD4^+^ T cells expressing PD-1. They also present high titers of anti-cardiac myosin and troponin autoantibodies in serum once the heart muscle is damaged, but not before the heart disease is clinically evident, suggesting that autoantibodies do not initiate the disease. In the same study, authors also demonstrate that bone marrow (BM) cells are sufficient to induce myocarditis. They created BM chimeras transferring PD-L1^−/−^; ^MRL+/+^ bone marrow cells into lethally irradiated WT; ^MRL+/+^ recipients and observed that PD-L1^−/−^ hematopoietic cells reproduce the disease although less severely. In those chimeras, they also observed a robust upregulation in the expression of PD-L1 in cardiac endothelial cells in the sites of inflammation [66].

Subsequent work has shown that PD-L1 expression not only in hematopoietic cells, but also in the heart, plays an important role limiting tissue damage once peripheral tolerance is compromised. In a mouse model of CD8^+^ T cell-mediated myocarditis, WT CTLs were transferred into PD-L1/2^−/−^ recipient mice and WT-bone marrow PD-L1/2^−/−^ chimeras, both models suffered lethal myocarditis with severe cardiac inflammation and cardiac immune infiltration compared with control WT recipients. Taking advantage of IFN-γ receptor-null mice they also demonstrated that PD-L1 overexpression in cardiac endothelial cells is dependent of IFN-γ signaling and it has a protective role limiting inflammatory damage to the myocardium [65].

#### 5.1.3. Ctla4 Knockout Mouse

CTLA-4 also has a role in maintaining peripheral tolerance to the heart. CTLA-4-deficient mice die prematurely due to lymphoproliferative disease with severe lymphocytic infiltration and tissue destruction in heart and pancreas [67]. Specific deletion of CTLA-4 in CD4^+^ Foxp3^+^ Treg cells was sufficient to induce the heart disease with cardiomegaly and mononuclear cardiac infiltration, although lifespan increased compared to full knockout animals [24]. Loss of CTLA-4 specifically in CD8^+^ T cells was also studied in a model of CD8^+^ T-lymphocyte-mediated myocarditis. Love et al. demonstrated that CTLA-4–deficient CTLs generated in the presence of IL-12 induced more severe disease characterized by cardiac cell infiltration and elevated lethality compared to control wild-type CTLs. In the same model, they also showed that CTLA-4–deficient CTLs, but not WT CTLs, generated without IL-12 also induced mild disease but less cell infiltration and less granzyme B production were observed [68].

#### 5.1.4. ICI Combination Knockout Models

In the clinic, different ICIs are prescribed in combination to improve therapeutic outcomes. Likewise, some genetic models of immune checkpoint double-knockout mice have been developed to study their phenotype when two immunomodulatory receptors with different functions are inhibited.

One of these is the mouse ICI-myocarditis model consisting of monoallelic loss of *Ctla4* and complete deletion of *Pdcd1* in C57BL/6 background*. Ctla4*^+/−^
*Pdcd1*^−/−^ mice die prematurely due to myocardial infiltration of T cells and macrophages, with higher mortality affecting females, although the life expectancy was higher compared to *Ctla4* full knockout animals. Cardiac phenotype was heterogeneous among animals and authors made an association between higher cardiac infiltration and cardiomegaly, ventricular wall thickening without systolic dysfunction and conduction disturbances. Cardiac disease improved in this model with a 12-week treatment with CTLA4-Ig starting at 21 days of age [69].

Another interesting model of combined immune checkpoint deletion is the double-knockout *Lag3*^−/−^ and *Pdcd1*^−/−^. Two groups reported independently that double deletion of *Lag3* and *Pdcd1* genes acts synergistically to cause lethal myocarditis while single deletion did not result in cardiac disease. The same cardiac phenotype was proven for BALB/c [70], C57BL/6 and B10.D2 backgrounds [71]. *Lag3*^−/−^ *Pdcd1*^−/−^ mice present hearts with massive infiltration of activated T cells and macrophages and have a life expectancy of less than 10 weeks but this improves when one of the genes is in heterozygosis, as the *Ctla4*^+/−^ *Pdcd1*^−/−^ model. Adoptive transfer experiments of T-cell-depleted splenocytes from double KO animals showed that both CD4^+^ and CD8^+^ population contributed to the disease. Authors concluded that the deletion of *Lag3* and *Pdcd1* caused the loss of peripheral tolerance in CD4^+^ and CD8^+^ T cells, that acquired an enhanced antigen-specific effector phenotype with increased expression of IFNγ and IL-17 [71].

### 5.2. Antibody Blockade of Immune Checkpoint Molecules Causes Myocarditis in Preclinical Models

All the models described in the previous section are genetic models in which complete inhibition of immune checkpoint molecules results in spontaneous myocarditis. However, the situation of ICI-derived myocarditis patients is very different, since cancer triggers an alteration of the immune system homeostasis and pharmacological inhibition of immune checkpoints is short-lived in patients’ lives. To address this issue, some new models have been developed in which tumor cells are present and ICI-inhibition is temporary.

#### 5.2.1. PD-1 Blockade

In vitro studies of nivolumab treatment in human embryonic-stem-cell-derived (hESC) cardiomyocytes have shown that inhibition of the PD-1/PD-L1 axis does not have a cardiotoxic effect by itself. However, the addition of nivolumab in co-cultures of hESC-derived cardiomyocytes with activated CD4^+^ T cells showed an increase in the expression of phosphorylated NFκB, phosphorylated STAT1, IFN-γ, and cleaved caspase-3 in the cardiomyocytes. Interestingly, the inflammatory and apoptotic response was weaker when nivolumab was added to cardiomyocytes and CD8^+^ T cell co-cultures [72]. In the same study, Tay et al. [72] treated melanoma tumor-bearing and non-tumor-bearing male BALB/cByJNarl mice with six doses of 250 μg of anti-PD1 every 72 h for 27 days. At the end of the treatment, they observed a reduced cardiac ejection fraction and fractional shortening, as well as increased CD4^+^ and CD8^+^ T cell infiltration in the myocardium in the anti-PD1 tumor-bearing group. In the same group, they also again found increased expression in the heart of phosphorylated NFκB, phosphorylated STAT1, IFN-γ and tumor necrosis factor (TNF)-α. However, they found no differences in the anti-PD1 non-tumor-bearing group compared to the tumor-bearing and non-tumor-bearing IgG treated mice, suggesting that the presence of cancer cells has an essential role in the development of ICI-myocarditis in this model.

In another model of melanoma tumor-bearing mice treated with anti-PD1, female C57BL/6N mice were injected with anti-PD1 in eight doses of 250 μg every other day for 12 days [73]. In this model, they observed an increased PD-L1 expression in endothelial cardiac cells and higher infiltration of CD4^+^ and activated CD8^+^ T cells in the myocardium of anti-PD1 treated mice. A moderate decrease in ejection fraction was also observed and left ventricular dysfunction was more severe in response to inotropic stress at the end of ICI treatment in tumor-bearing mice. The investigators linked systolic dysfunction to changes in cardiac lipid metabolism in anti-PD treated mice. As in the previous model, cardiac dysfunction was not reproduced in mice receiving anti-PD1 and without tumors. Additionally, CD8^+^ T cell depletion and TNFα blockade was tested to prevent ICI-cardiotoxicity in this model. Both therapies preserved left ventricle function in the anti-PD1 treated mice, however CD8^+^ T cell depletion abolished the anti-tumor activity of the ICI. Conversely, anti-TNFα preserved anti-cancer efficacy, showing potential as therapy for ICI-induced cardiotoxicity. However, it must be noted that in the clinical setting the use of TNF-α blocking drugs is contraindicated in patients with symptoms of heart failure, limiting the translation of this therapy (New York Heart Association class III, IV).

#### 5.2.2. CTLA-4 Blockade

IL-17-producing T cells are a pivotal axis in the pathophysiology of myocarditis in human and mouse models, initiating inflammation and also the progression of acute myocarditis to dilated cardiomyopathy (DCM) [48,74,75,76]. Antibodies blocking CTLA-4-B7 interaction have been shown to potentiate Th17 differentiation in vitro and in vivo and Th17-mediated autoimmunity in murine models [77]. In a model of EAM, the treatment of WT BALB/c mice with anti-CTLA-4 antibodies significantly exacerbated the severity of the disease. Hearts of anti-CTLA-4 treated mice presented higher infiltration of CD3^+^ T cell that where mainly IL-17^+^ CD4 T cells [77]. Additionally, it was observed that CTLA-4 blockade increases Th17 cell number in peripheral blood of metastatic melanoma patients [78].

#### 5.2.3. ICI Combination Blockade

Despite their heterogeneity, most reports indicate that the use of ICI in combination increases the risk of developing ICI myocarditis. Some animal models were established to study myocarditis after combination of different ICI blocks and the results seem to support this observation.

In a murine model of lung metastatic colon cancer, BALB/c tumor-bearing mice were injected with anti-PD1 and anti-PDL1, either sequentially or simultaneously administered. Animals treated with combination therapy, but not those treated with anti-PD-1 or anti-PD-L1 alone, presented myocyte injury and mononuclear infiltrates in the myocardium consisting mainly of neutrophils and inflammatory monocytes [79].

In another model of humanized C57BL/6 mice expressing the human CTLA-4 protein, perinatal mice that were treated with anti-PD1 plus ipilimumab showed severe dilated cardiomyopathy and lymphocyte infiltration of CD3^+^ T cells. The myocarditis lesions appeared as well in adult CTLA-4 humanized MC38-tumor-bearing mice treated with ipilimumab alone or in combination with anti-PD1. In this combination therapy model, although prompting similar efficacy to monotherapy eliminating the tumor, it led to increased toxicity to the heart [80]. Nivolumab plus ipilimumab was also administered weekly for one month to cynomolgus monkeys. Investigators observed multiple organ toxicities, including myocarditis, and increased proliferation of T cells, activated and memory T cells, and cytokine production in peripheral blood. Myocarditis lesions were composed of mainly T cells, with more CD4^+^ than CD8^+^ T cells, and macrophages. RNA-sequencing analysis of heart tissues from combination ICI-treated monkeys revealed dysregulation of immune pathways in the heart mainly involved in Th cell differentiation, co-stimulation, proliferation, and cytokine production [81].

## 6. Recent Insights into Human ICI-Associated Myocarditis Development

Altogether, clinical trials and studies in humans and animal models are helping to gather all the pieces to shed some light on the molecular mechanisms underlying ICI-myocarditis development. Multiple mechanisms could be contributing in promoting the anti-heart immune response observed in this pathology.

Lv et al. revealed the presence of T cells in the periphery reactive to the MyHCα and demonstrated that the cause is a lack of representation of this antigen in the mTECs of mice and humans [82]. This breach of self-tolerance could explain the existence of peripheral T cells specific for cardiac antigens that drive autoimmunity upon ICI-induced unspecific de-repression. Incidentally, ICI administration has been proved to increase the diversity of the TCR repertoire [83], and have a direct correlation between this factor and the development of severe irAEs in melanoma patients [84]. This links ICI treatment not only with the de-repression of existing autoreactive T cells but also with an increase in the pool of T cell recognizing antigens that would prompt autoimmune responses. Further, the inhibition of the PD-1/PD-L1 axis impairs one of the mechanisms of the heart to maintain its peripheral tolerance to autoreactive T cells clones through the over-expression of PD-L1, which has been observed in myocarditis affected areas of human cardiac biopsies [37].

Another mechanism suggested to contribute to the development of toxicity is the release of potential antigens from the tumor that resemble cardiac or muscle antigens directing the immune response towards the myocardium, especially in the case of melanoma patients. This was proposed after the observation of myocardial T cell infiltration and clonal expansion of T cells with shared receptors in the heart and tumor in two melanoma patients. Both patients suffered myositis as well, and clonal T-cell populations infiltrating myocardium and tumor were also present in skeletal muscle [37]. This is further supported by the fact that myositis and myasthenia gravis are the most common concomitant irAEs that occur with myocarditis [11]. Further, the odds of developing myocarditis are different depending on the type of cancer being treated, with melanoma patients experiencing the higher risk [49] as reported up to the date of this review. This may underscore the hypothesis that antigens released by different tumors develop varied adverse responses. Additionally, tumor-free ICI-treated mice did not develop cardiotoxicity in two models of ICI-myocarditis following anti-PD1 treatment in melanoma tumor-bearing mice [70,71], providing further evidence that tumor prompt this pathology.

Another noteworthy hypothesis is the relevance of cytokines in disease induction through tissue damage. Based on knowledge derived from studies in mice, Th17 cells have been reported to be a key population infiltrating the heart in ICI-associated myocarditis [77]. As already stated, IL-17 signaling has a major role during acute myocarditis initiating inflammation and progression to DCM. CD4^+^ IL-17^+^ cells have been found to be increased in peripheral blood from acute myocarditis patients [48,76] but also in patients with melanoma under ICI treatment [78]. One study associated baseline circulating IL-17 with the posterior development of severe colitis [85], and another recent study underpins the tight link between Th17 individual signature and the development of ICI-associated myocarditis [86].

These mechanisms are not necessarily mutually exclusive, as autoreactive T cells, antigen cross-reactivity, and cytokine-induced tissue damage could be inciting factors contributing to myocarditis induction. Overall, the presence of the tumor and ICI-induced enhancement of the immune response, could explain the increased risk of developing myocarditis observed in such patients compared with the incidence in the general population [54].

All these processes together seem to drive an autoimmune response against the heart which is characterized by the myocardial infiltration of T cells, mainly alongside with macrophages. T cell infiltrate is composed of CTL and Th effector cells, the last being more prominent [87]. These observations in human cardiac biopsies are supported by the immunophenotype of cardiac infiltration observed in preclinical models of ICI-myocarditis, expounded in the previous section of this review.

With the current information of interactions between each cellular player and the kinetics of the immune response, some mathematical models have emerged to predict disease progress [88]. Modelling will be further improved as a consequence of deeper characterization of pathogenesis mechanisms. Moreover, few of the reported cases manage to overcome the tumor, myocarditis, and/or other adverse effects. Hence, there is a gap in the knowledge of the effects that long-term exposure to ICIs can induce in the cardiovascular system, which is necessary to fully understand the effect of this therapy.

## 7. Conclusions and Future Directions

ICI-myocarditis is a rare complication among all irAEs and among cardiac events that occur as a consequence of the de-repression of mechanisms containing T cell activity. There is great variability between studies regarding the prevalence of ICI-myocarditis. Therefore, the search for preliminary markers that allow us to identify the population at risk is a very active field.

The mechanisms that induce this pathology are also a current field of research since inflammation often leads to heart failure and other major cardiac events that carry a high mortality rate. For all these reasons, the scientific community has developed several disease models that reflect the particular etiology of this type of myocarditis. Three mechanisms have been identified that explain the development of this cardiomyopathy, and which can occur in isolation or together causing aggravation; (1) the autoimmune response against the myocardium could be caused by a breakdown of peripheral tolerance that is enhanced by ICI treatment, (2) cross-reactivity of tumor antigens that resemble cardiac or muscle antigens, and (3) elevated blood levels of IL-17a triggered by ICI treatment that have been linked to tissue damage and increased risk of developing ICI-myocarditis.

A wider comprehension of the molecular pathways involved in this disease could highlight targets of interest for the development of (1) sensitive, specific and feasible diagnostic tools able to detect subclinical presentations and (2) more tailored and personalized therapeutic approaches that can prevent the aggravation and the development of the disease without the need for ICI discontinuation.

## Figures and Tables

**Figure 1 cancers-14-04494-f001:**
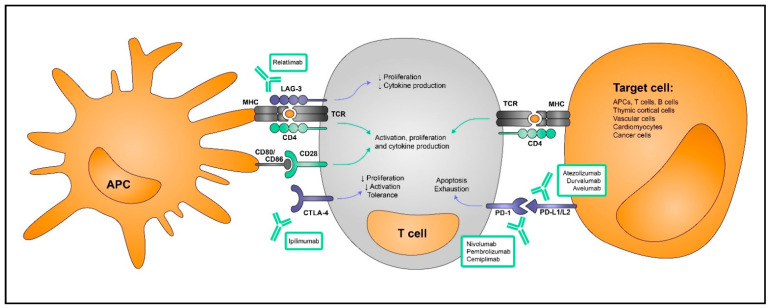
Immune checkpoint molecules. T cells become active upon binding of the TCR to the MHC-antigen complex followed by CD28 co-stimulation both provided by APCs. Both LAG-3 and CTLA-4 limit these interactions via direct competition, hence rendering T cell activation and subsequent proliferation and cytokine production. On the other hand, PD-1 and PD-L1/L2 binding on the target cells also contribute to limit the immune response by induction of apoptosis and exhaustion on T cells. These are also targeted by monoclonal antibodies to enhance immune responses in a cancer setting.

**Table 1 cancers-14-04494-t001:** Approved ICI therapies by the Federal Drug Administration (FDA).

Immune Checkpoint	Immune Checkpoint Inhibitor	FDA Approval Date	Approved Target Cancers
CTLA-4	Ipilimumab	2011	Unresectable and metastatic melanoma (alone, with nivolumab or as an adjuvant) Advanced renal cell carcinoma (in combination with nivolumab) Microsatellite instability-high or mismatch repair deficient metastatic colorectal cancer (in combination with nivolumab) Hepatocellular carcinoma (alone or in combination with nivolumab) Metastatic non-small cell lung cancer (in combination with nivolumab) Unresectable malignant pleural mesothelioma (in combination with nivolumab) Unresectable and metastatic esophageal cancer (in combination with nivolumab)
PD-1	Nivolumab	2014	Unresectable and metastatic melanoma (alone, with ipilimumab or with relatlimab) Resectable non-small cell lung cancer (as a neoadjuvant) Metastatic non-small cell lung cancer (alone or in combination with ipilimumab) Malignant pleural mesothelioma (in combination with ipilimumab) Advanced renal cell carcinoma (alone or with ipilimumab) Refractory classical Hodgkin lymphoma Recurrent and metastatic squamous cell carcinoma of the head and neck Advanced and metastatic urothelial carcinoma Microsatellite instability-high or mismatch repair deficient metastatic colorectal cancer (in combination with ipilimumab) Hepatocellular carcinoma (alone or in combination with ipilimumab) Unresectable and metastatic esophageal cancer (alone or in combination with ipilimumab) Resected esophageal cancer (as an adjuvant) Advanced and metastatic gastric cancer, gastresophageal junction cancer, and esophageal adenocarcinoma
	Pembrolizumab	2014	Unresectable and metastatic melanoma Stage III and metastatic non-small cell lung cancer Unresectable and metastatic head and neck squamous cell cancer Refractory classical Hodgkin lymphoma Refractory primary mediastinal large B cell lymphoma Advanced and metastatic urothelial carcinoma Microsatellite instability-high or mismatch repair deficient solid tumors Microsatellite instability-high or mismatch repair deficient metastatic colorectal cancer Advanced unresectable and metastatic gastric cancer Unresectable and metastatic esophageal cancer Recurrent and metastatic cervical cancer Hepatocellular carcinoma Advanced and metastatic Merkel cell carcinoma Renal cell carcinoma Endometrial carcinoma Unresectable and metastatic tumor mutational burden-high cancer Recurrent and metastatic cutaneous squamous cell carcinoma Recurrent unresectable and metastatic triple-negative breast cancer
	Cemiplimab	2019	Metastatic cutaneous squamous cell carcinoma Advanced and metastatic basal cell carcinoma Advanced and metastatic non-small cell lung cancer
PD-L1	Atezolizumab	2016	Advanced and metastatic urothelial Metastatic non-small cell lung cancer Small cell lung cancer Unresectable and metastatic hepatocellular carcinoma Unresectable and metastatic melanoma
	Durvalumab	2017	Advanced and metastatic urothelial carcinoma Stage III non-small cell lung cancer Advanced small cell lung cancer
	Avelumab	2017	Metastatic Merkel cell carcinoma Advanced and metastatic urothelial carcinoma Advanced renal cell carcinoma
LAG-3	Relatlimab	2022	Unresectable or metastatic melanoma (in combination with nivolumab)

## Data Availability

Not applicable.

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
