# Peer review of "Pathophysiology of Immune Checkpoint Inhibitor-Induced Myocarditis"

_cancers, 2022, doi:10.3390/cancers14184494_

Round 1
Reviewer 1 Report
You offer an important topic for discussion. The general remark is that the review is descriptive rather than analytical.
I have also some particular observations
1. Are there Immune Checkpoint receptors on cells originated from non-immune tissues?
2. Please explain clearer the hypothesis and evidence regarding the pathophysiology of toxicity against healthy tissues and, in particular, cardiotoxicity and myocarditis caused by ICIs? Do these complications develop due to the expression of immune checkpoints molecules on non-immune cells, or due to inappropriate activation of autoimmune T cell clones, or there are some other reasons?
3. What pathologies / conditions can be considered as risk factors for the development of myocarditis during ICI treatment?
4. The authors describe the mechanisms of immune tolerance in the chapter 2. In this regard, it would be appropriate to describe in the same chapter the mechanisms and signaling pathways of correspondent axes as PD-1/PD-1L/L2, CTLA-4/CD80/86, and their role in normal physiology, and not in chapter “Immune checkpoint inhibitors in cancer”. Graphical image would be very appropriate, also indicating the targets for particular ICIs.
5. Why do adverse effects depend on the type of ICIs? For instance, as it is mentioned in MS, why colitis in patients treated with anti-CTLA-4 is more frequent, while pneumonitis, hepatitis and neurotoxicity are the main cause of death in patients in whom the PD-1/PD-L1 axis is inhibited?
6. How does Table 1 relate to the main topic of the review?
7. In Table 2, the column "authors" is not needed, the column "references" is sufficient.
8. The percentages of ICI-associated myocarditis reported by different authors (Table 2) varied significantly: from 0.0003 % (Jain et al., 2021) to 100% (Moslehi et al., 2018; Zhang et al., 2020; Champion et al., 2020; Puzanov et al., 2021). Please discuss this point.
9. What is the treatment proposed for ICI-associated myocarditis? Are there any cardioprotectors that could be recommended to be used simultaneously with ICIs, to prevent cardiotoxicity?
10. The title of chapter 5.1. is formulated as “Preclinical models of myocarditis by genetic deletion of Immune Checkpoint molecules”. May be the title as “Genetic deletion of Immune Checkpoint molecules causes myocarditis in preclinical models” would reflects more precisely the main message if this chapter? The same observation for the title of 5.2 chapter: “Antibody blockade of Immune Checkpoint molecules causes myocarditis in preclinical models” instead of “Preclinical models of myocarditis by antibody blockade of Immune Checkpoint molecules”. It should be noted that it is these models that should be given special attention in order to draw important conclusions about the mechanisms of ICI-associated myocarditis.
11. The conclusions, as they are formulated, look very vague. The authors should draw more definite conclusions of mechanisms of ICI-associated myocarditis from critical analysis of their proposed topic. If they cannot offer more definite conclusions, the publication of this article is rather premature.
12. English style should be corrected.
Reviewer 2 Report
This is an exciting review discussing the potential pathophysiology of ICI-induced myocarditis. The authors did an outstanding job summarizing the literature, including pre-clinical data. In regards to flow and style, the article reads well.
The comments below can be considered to make the review even stronger:
· I suggest including in the discussion the role of troponin in the diagnosis of ICI-myocarditis. Is there a threshold value above which the diagnosis is more likely?
· I suggest discussing the controversy regarding the optimal dose of steroids to be used when myocarditis is suspected – 1000mg vs. 1-2mg/kg.
Minor comments
- Please include references to data cited on lines 188-190.
- There is a typo “evicing” in line 235 – should it be “evidencing”?
- De-repression is spelled both with and without an – throughout the manuscript. I suggest choosing only one option.
Round 2
Reviewer 1 Report
he article has been improved and can be accepted for publication.